# Analysis of lncRNAs and Their Regulatory Network in Skeletal Muscle Development of the Yangtze River Delta White Goat

**DOI:** 10.3390/ani14213125

**Published:** 2024-10-30

**Authors:** Wenjun Tang, Jiahao Sun, Rahmani Mohammad Malyar, Fangxiong Shi

**Affiliations:** 1College of Animal Science and Technology, Nanjing Agricultural University, Nanjing 210095, China; 2022105018@stu.njau.edu.cn (W.T.); 2022205014@stu.njau.edu.cn (J.S.); 2021205039@stu.njau.edu.cn (R.M.M.); 2Lishui Institute of Agriculture and Forestry Sciences, Lishui 323000, China; 3Veterinary Science Faculty, Nangarhar University, Nangarhar 2601, Afghanistan

**Keywords:** Yangtze River Delta White (YDW) goat, lncRNAs, *M. Longissimus dorsi*, meat quality, muscle fiber characteristics, ceRNA network

## Abstract

For most mammals, lncRNA plays an important role in normal physiological activities and development. At present, studies on the growth and development of goats through lncRNAs have been performed, but there are still limited studies on some native goat breeds of China, and studies on the dynamic change process of lncRNAs are also very scarce. In this study, we identified potential new lncRNAs, described lncRNA expression profiles at different developmental stages of the Yangtze River Delta White (YDW) goat, and constructed corresponding ceRNA regulatory networks. The results provide valuable genomic data for studying the growth and development process of animals, and they may provide a theoretical basis for studying the biological functions of lncRNAs in animal growth and development.

## 1. Introduction

For a long time in the past, it was believed that gene regulation was primarily the function of protein-coding genes, and that non-coding RNAs were useless. However, with the development of transcriptome sequencing technology, more and more non-coding RNAs, including lncRNAs, circRNAs (circular RNA), and miRNAs (microRNA) were unearthed, and it was found that these so-called non-coding RNAs might play an important role in protein-coding gene expression and its regulation [1]. It has been shown that although approximately two-thirds of the mammalian genome is actively transcribed, less than three percent of the proteins are coded for, and a large part of this transcriptional property is regulated by lncRNAs [2,3]. Long non-coding RNA is an RNA transcript larger than 200 nt in length and mostly with no coding ability [4]. This does not have open reading frames, is expressed in a more tissue-specific manner than mRNAs, and is less conserved across species [5,6]. Currently, the important roles played by lncRNAs include regulation of transcription [7], regulation of the cell cycle and apoptosis [8], influence on embryogenesis and organ differentiation [9], epigenetic modification, and selective shearing [10,11]. However, among the identified lncRNAs, only a few have been functionally annotated, and the functions of many lncRNAs remain unclear.

Skeletal muscle is composed of a large number of muscle cells arranged in bundles with the ability to contract. It is the largest tissue in the body of livestock and generally includes myomeres, myofibrils, muscle fibers, and muscle bundles, which together form the muscle [12,13]. Skeletal muscle development is mainly determined by the proliferation and differentiation of myofibroblasts, including three main processes: myogenesis, fibrosis, and adipogenesis [14,15]. Previous studies have shown that the growth and development of skeletal muscle in mammals are different before and after birth: the number of muscle fibers increases before birth, while the hypertrophy and transformation of muscle fibers after birth are prominent features of skeletal muscle development [16,17]. Skeletal muscle growth and development directly affect meat production performance, meat quality, and other traits that are closely related to economic benefits. In addition to the functional genes identified in previous studies, such as members of the muscle regulatory factor (MRF) family and insulin-like growth factor (IGF) family, the development of skeletal muscle in livestock and poultry is regulated by long non-coding RNA, which is also complex in its regulation [18]. Annotating the expression and function of coding genes and non-coding RNAs to explore the mechanisms of skeletal muscle growth and development is still an important methodology [19,20].

The Yangtze River Delta White (YDW) goat is an excellent native goat breed in China, which has excellent characteristics such as low odor, tender meat, and an even fat distribution, and it is also the only goat breed in China that can produce high-quality penned wool, with excellent skin, meat, and wool. In this study, we used high-throughput sequencing technologies to analyze the transcriptome of the longissimus dorsi of YDW goats from different periods, to understand the dynamic change process of lncRNAs in the skeletal muscles of goats, to predict the potential functions of lncRNAs, and to construct ceRNA networks related to the analyzed lncRNAs. These results can help in understanding the differences in muscle fiber development and muscle quality of YDW goats at different developmental stages, as well as the molecular mechanisms that regulate skeletal muscle development throughout the life cycle.

## 2. Materials and Methods

### 2.1. Animals and Sample Preparation

The 9 healthy YDW male goats involved in this experiment were all from the goat breeding farm in Lishui City, Zhejiang Province, China. Three goats each at 2, 6, and 10 months of age were selected; the goats of the same age were similar in weight and body condition, and the feeding management and nutrition levels of the goats were the same. After slaughter, longissimus dorsi samples were collected between the 12th and 13th thoracic vertebrae of each goat, and they were immediately placed in liquid nitrogen for rapid freezing and transferred to a −80 °C refrigerator for storage. Some of the longissimus dorsi samples were soaked in 4% paraformaldehyde solution for the observation of muscle fiber characteristics, and other samples were refrigerated at 4 °C for the determination of meat quality indexes.

### 2.2. Observation of Muscle Tissue Sections

The collected samples of *M. Longissimus dorsi* were immersed and fixed in 4% paraformaldehyde, and after dehydration with alcohol and transparency with xylene, the samples were sectioned at a thickness of approximately 5 μm using paraffin embedding. Tissue sections were stained using the hematoxylin–eosin method, and images were collected for the determination of muscle fiber diameter and cross-sectional area using Image Pro Plus 32.0 (Media Cybernetics Inc., Rockville, MD, USA) software.

### 2.3. Determination of Meat Quality

#### 2.3.1. pH Value

Referring to the determination method of Brant [21], a pH meter (testo-pH2, Germany) was used to determine the pH value of goat muscle at 45 min (pH_45min_) and 24 h (pH_24h_) after slaughter (pH value of meat samples stored at 0~4 °C for 24 h). Before the measurement, a standard solution with pH = 4.00 and pH = 7.00 was used for calibration. Each sample was measured at different locations five times, and the average value was taken for statistical analysis.

#### 2.3.2. Meat Color

Referring to the measurement method of Zhao [22], which was slightly modified, the meat color of goat muscle in three periods was determined by a colorimeter (CR8, Three Technology Co., Ltd., Shenzhen, China). Before the determination, the colorimeter was calibrated using a standard whiteboard, and the meat color was measured at three different muscle surface positions; the results were averaged. The results were expressed as lightness L*, redness a*, and yellowness b*.

#### 2.3.3. Shear Force

Referring to the determination method of Honikel [23], the muscle samples were placed in self-sealing bags and heated in a water bath until the internal temperature of the samples reached about 70 °C. The samples were removed, cooled to room temperature, and then the shear force was determined by using a shear force meter (Bulader Technology Development., Ltd, Beijing, China) perpendicular to the direction of the muscle fibers. The measurement was repeated three times for each sample.

#### 2.3.4. Drip Loss

Referring to the determination method of Li [24] with slight modifications, muscle samples were trimmed into long strips of 2 × 3 × 5 cm, weighed and suspended in a sealed self-sealing bag using a wire without making contact with the bag, and placed in a refrigerator at 4 °C for 24 h. The meat samples were removed and weighed by absorbing the surface water with a filter paper, and the drip loss was calculated as the ratio of the lost weight to the initial weight.

### 2.4. Extraction of Total RNA and Construction of Library

Total RNA was extracted from longissimus dorsi of goats by the Trizol (Tiangen, Beijing, China) method. The concentration and integrity of RNA were detected by an Agilent 5400 (Agilent Technologies, Palo Alto, CA, USA) analyzer. Agarose gel electrophoresis was used to detect the purity of RNA, and the samples were qualified for library construction and transcriptome sequencing after passing the detection. Sequencing libraries were generated using the NEBNext Ultra Directional RNA Library Prep Kit for Illumina (NEB E7420, Boston, MA, USA) according to the manufacturer’s recommendations. First, rRNA was removed using the TruSeq Stranded Total RNA Ribosome Removal Kit (Life Technologies, Carlsbad, CA, USA), which maximizes the retention of lncRNAs containing polyA tails. We added fragmentation buffer to the enriched RNA to break the RNA into small fragments. Then, the first strand of cDNA was synthesized by reverse transcription based on the fragmented RNA as a template, and the second strand of cDNA was synthesized by the addition of buffer, dNTPs, DNA polymerase I, and RNase H. After purification, terminal repair, addition of A, and connection of a sequencing joint, the second strand of cDNA containing U was degraded by the USER enzyme and enriched by PCR. Finally, the PCR products were purified by AMPure XP beads (Beckman Coulter, Beverly, MA, USA), to obtain specific mRNA and lncRNA libraries. The construction of a Small RNA library needs to be carried out individually, taking advantage of the special phosphate group and hydroxyl structure at the 3′ and 5′ ends of Small RNA, using total RNA as the starting sample, directly adding junctions at both ends of the Small RNA, synthesizing cDNA by reverse transcription, followed by PCR amplification, and then separating the target DNA fragments through PAGE gel electrophoresis. The cDNA library was obtained after cutting and recycling. Finally, all sequencing libraries were sequenced by Illumina NovaSeq 6000 (Illumina, San Diego, CA, USA) after being qualified by a sequencing company (Novogene, Beijing, China).

### 2.5. Quality Control and Comparison of Reference Genomes

Raw reads in the fastq format are processed using internal perl scripts that prune reads containing adapters, ploy-N, or low-quality reads to obtain clean data, while calculating the Q20, Q30, and GC contents of the clean data. The index of the reference genome was constructed using HISAT2 (version 2.0.5), and the paired-end clean reads were compared with the goat reference genome. Stringtie (version 1.3.3) was used to calculate the number of reads mapped to each gene, and FPKM (number of fragments per kilobase of transcript sequence per million base pairs sequenced) was calculated to quantify the size of the genome.

### 2.6. Identification and Differential Expression Analysis of lncRNAs

The Stringtie (version 1.3.3) software was used to merge the transcripts obtained from the concatenated samples, and the transcripts with an uncertain chain direction or a transcript length less than 200 nt were removed. The transcripts were compared with known databases by gffcompare (version 0.10.6) software, and the known transcripts in the databases were filtered out. Finally, the intersection of transcripts without coding potential was selected from the three database results, of CPC2 [25], PFAM [26], and CNCI [27], as the candidate novel lncRNA dataset for our prediction. By referring to HGNC [28] (The HUGO Gene Nomenclature Committee) for the naming criteria of long non-coding RNAs, the candidate novel lncRNAs were screened and named according to their positional relationship with the coding gene. For each sequencing library, the edgeR software package (version 3.22.5) was used to perform differential expression analyses under two conditions before combined differential expression analysis. The significantly differentially expressed genes or lncRNAs were screened with an adjusted *p*-value < 0.05 and |log2FoldChange| ≥ 1 as the conditions. The smaller the obtained adjust *p*-value, the more significant the difference.

### 2.7. Prediction of Differentially Expressed lncRNA Target Genes and Enrichment Analysis of GO and KEGG

In order to understand the function of lncRNAs, the function of differential lncRNA target genes is predicted first. According to the mode of action of transcripts on target genes, they are classified as cis-regulatory target genes (within 100 kb upstream and downstream of differentially expressed lncRNAs) and trans-regulatory target genes of lncRNAs (correlation of differentially expressed lncRNAs with the expression of protein-coding genes). GO and KEGG enrichment analyses of differentially expressed genes were performed by cluster profiler (version 3.8.1) software. GO term analysis screened differentially expressed genes corresponding to biological functions, and KEGG pathway enrichment analysis identified metabolic pathways or signal transduction pathways significantly enriched in differentially expressed genes. We focused on pathways with *p* < 0.05.

### 2.8. Construction of lncRNA–miRNA–mRNA ceRNA Networks

The differentially expressed RNA molecules between the various groups were analyzed for targeting relationships. First, the intersection analysis of differentially expressed lncRNA target genes and mRNAs suggested that the intersection part of lncRNAs and mRNAs might have a relationship of targeted regulation. Secondly, the target lncRNAs of differentially expressed miRNAs were predicted by miRanda software. Then, the intersection of miRanda (version 1.2) and RNAhybrid (version 2.1.2) software was taken as the target gene prediction of miRNA. According to ceRNA theory, there is a competitive relationship between interacting miRNA and lncRNA, so the expression level is negatively correlated. According to the principle of miRNA inhibition on mRNA, miRNAs and mRNAs with a negative correlation in expression level were screened as target gene pairs. Pearson’s correlation coefficient (PCC < −0.85) was used as the condition to evaluate the negative correlation between miRNA and lncRNA, miRNA, and mRNA. The ceRNA regulatory network constructed with the above data was visualized using Cytoscape (version 3.7.2) software.

### 2.9. Quantitative Real-Time PCR (qPCR) Analysis

Total RNA was extracted from *m. longissimus dorsi* of goats by Trizol (Tiangen, Beijing, China). RNA was reversed transcribed into cDNA by HiScript III RT SuperMix for qPCR (Vazyme, Nanjing, China). qPCR was performed using 2 × PerfectStart Green qPCR SuperMix (TRAN, Beijing, China). The 20 μL system consists of 0.4 μL forward primer and 0.4 μL reverse primer, 2 × PerfectStart Green qPCR SuperMix 10 μL, template cDNA 1 μL, and nuclease-free water 8.2 μL. The following reaction was performed on the ABI QuantStudio5 (Thermo, Shanghai, China) system: stage 1, 94 °C for 30 s; stage 2 (45 PCR cycles), 94 °C for 5 s, 60 °C for 15 s, 72 °C for 10 s. The primer sequence of qPCR is shown in Table 1. The results were statistically analyzed by the 2^−ΔΔCT^ method [29].

### 2.10. Statistical Analysis

The experimental data were sorted by Excel2016 and statistically analyzed by SPSS 26.0 (SPSS Inc., Chicago, IL, USA), and GraphPad Prism 8 (GraphPad Prism Ink, Boston, MA, USA) was used to draw graphs. The statistical results were expressed as the mean ± SEM. One-way analysis of variance was used to compare the data of multiple groups, and the results were statistically significant with *p* < 0.05.

## 3. Results

### 3.1. Observation of Muscle Fiber Characteristics in Goats of Different Ages

HE (hematoxylin–eosin staining) sections of the *m. longissimus dorsi* revealed differences in the muscle fibers of YDW goats at different ages (Figure 1a–c). The muscle fiber diameter and muscle fiber area of 6-month-old and 10-month-old goats were significantly greater than those of 2-month-old goats (*p* < 0.01), with no significant difference observed between the 6-month-old and 10-month-old goats (*p* > 0.05) (Figure 1d,e).

### 3.2. Quality Comparison of Goat Meat of Different Ages

The meat quality of YDW goats at 2, 6, and 10 months of age is shown in Table 2. a* and b* values of the muscles of goats at 6 months of age were significantly greater than those of goats at 2 and 10 months of age (*p* < 0.01), and L* values of the muscles of goats at 2 and 6 months of age were significantly greater than those of goats at 10 months of age (*p* < 0.01). The pH_45min_ of 10-month-old goat muscle was higher than that of 2-month-old goats (*p* = 0.006) (Figure 2b). After the meat samples were stored at 4 °C for 24 h, the pH_24h_ of the muscles of 10-month-old goats was significantly lower than that of 2-month-old and 6-month-old goats (*p* < 0.01) (Figure 2c), and the pH values of the muscles of goats decreased in all groups, where the greatest decrease in pH was observed in 10-month-old goats compared to 2-month-old and 6-month-old goats. The shear force of the muscle was the smallest at 2 months of age and increased progressively as the age of the goats increased (*p* < 0.05) (Figure 2a). The drip losses of goats at 2, 6, and 10 months of age were 7.8%, 12%, and 9.5%, respectively, and there were no significant differences between the groups (*p* > 0.05) (Figure 2d).

### 3.3. Quality Control and Comparative Analysis of Sequencing Data

The libraries of 9 goats at different growth stages were constructed and named M2-1, M2-2, M2-3, M6-1, M6-2, M6-3, M10-1, M10-2, and M10-3, respectively. After Illumina sequencing, raw reads were obtained, and after filtering the raw sequencing data, detecting the incidence of sequencing errors and the distribution of GC content, the clean reads of each goat sample were distributed in the range of 80757046 (M10-1) to 89638430 (M2-1), the Q20 of each sample was larger than 97%, and Q30 was larger than 94%. When using Hisat2 (http://ccb.jhu.edu/software/hisat2 accessed on 10 April 2024) software to give clean reads for a comparison with the goat reference genome (https://www.ncbi.nlm.nih.gov/datasets/genome/GCF_001704415.2/ accessed on 10 April 2024), it was found that more than 94% of the clean reads were matched with the reference genome in each sample (see Table 3 for detailed sequencing results). In summary, the sequencing results were accurate and reliable, making them suitable for subsequent analysis.

### 3.4. Screening and Identification of lncRNA

The final number of known lncRNAs identified in this study was 2676, and the number of predicted new lncRNAs was 397, of which 1777 lncRNAs were expressed in all three periods of time in goats, and 110, 93, and 99 lncRNAs were specifically expressed in 2-month-old, 6-month-old, and 10-month-old goats (Figure 3d). The screened lncRNAs were, respectively, classified into four types: lincRNA (37.53%), antisense lncRNA (19.90%), overlapping lncRNA (29.72%), and intronic lncRNA (12.85%) (Figure 3e). In order to obtain the differences between lncRNAs and mRNAs and verify whether the predicted novel lncRNA conformed to the general characteristics, lncRNA and mRNA were compared and analyzed for three aspects: transcript length, transcript exon number, and ORF length (open reading frame) (Figure 3a–c). It was found that the average transcript length, average transcript exon number, and ORF length of novel lncRNAs were 4353, 2.6, and 178, respectively, which were smaller than those of novel mRNAs with an average transcript length of 63,062, average transcript exon number of 2.8, and average ORF length of 363. In addition, we calculated the distribution of expression values and expression levels of the transcripts in each sample (Figure 3f).

### 3.5. Analysis of Differentially Expressed mRNA and lncRNA

The number of differentially expressed mRNAs and lncRNAs between different growth stages was determined. It was found that the numbers of differentially expressed mRNAs were 1028, 1099, and 740 and the numbers of differentially expressed lncRNAs were 92, 72, and 68 at the ages of 2 months and 6 months, 2 months and 10 months, and 6 months and 10 months, respectively (Figure 4a,b). A differential volcano map shows the top ten significantly up- and down-regulated lncRNAs in order of significance. We also performed a pairwise comparison of group classifications (Figure 4c). In order to reveal the differential expression patterns of lncRNAs in goat muscles during the three periods, hierarchical cluster analysis was performed on the differentially expressed lncRNAs (Figure 4d).

### 3.6. Prediction and Enrichment Analysis of lncRNA Target Genes

In order to predict the functions of lncRNAs, the target genes of lncRNAs were predicted by the positional relationship (co-location) and expression correlation (co-expression) between lncRNAs and protein-coding genes, and it was found that a total of 433 genes may be cis-targeted and regulated by lncRNAs, GO functional enrichment, and KEGG pathway enrichment of target genes. It was found that a total of 51 GO terms were significantly enriched (*p* < 0.05), and the top 10 GO terms in order of significance included protein heterodimerization activity, protein dimerization activity, chemokine activity, and chemokine receptor binding, suggesting that lncRNAs may affect muscle growth and development by regulating the activity of adjacent proteins and chemokine binding. Meanwhile, the results of KEGG pathway analysis showed that insulin secretion and the thyroid hormone signaling pathway were significantly enriched. The trans-targeted regulation of lncRNA was analyzed, and a total of 1378 lncRNA transcripts and 24,987 protein-coding transcripts were found to have co-expression interactions. Functional analysis showed that the co-expressed genes were significantly enriched for 100 GO terms, including the protein metabolic process, gene expression, cellular catabolic process, and chemokine receptor binding. In addition, the significantly enriched KEGG pathways included the NF-kappa B signaling pathway, insulin signaling pathway, and other pathways related to muscle growth and development.

### 3.7. Construction of lncRNA–miRNA–mRNA ceRNA Network

The construction of the ceRNA network is based on the molecular sponge action of lncRNA, which reduces the effect of miRNA on target genes through competitive binding of miRNA, thus regulating the relative expression of target genes [30]. In this study, the ceRNA network of 2-month-old and 6-month-old goats included 53 miRNAs, 71 lnRNAs, and 239 mRNAs (Figure 5a), among which there were 42 up-regulated and 29 down-regulated lncRNAs, 34 up-regulated miRNAs and 19 down-regulated miRNAs, and 113 up-regulated mRNAs and 126 down-regulated mRNAs. The ceRNA network of 2-month-old and 10-month-old goats included 20 miRNAs, 47 lncRNAs, and 80 mRNAs (Figure 5b). Among those, there were 7 up-regulated and 13 down-regulated miRNAs, 27 up-regulated and 20 down-regulated lncRNAs, and 26 up-regulated and 54 down-regulated mRNAs. The ceRNA network of 6-month-old and 10-month-old goats included 46 miRNAs, 58 lncRNAs, and 162 mRNAs (Figure 5c), among which there were 12 up-regulated miRNAs and 34 down-regulated miRNAs, 34 up-regulated lncRNAs and 24 down-regulated lncRNAs, and 54 up-regulated mRNAs and 108 down-regulated mRNAs. It is worth noting that some lncRNA–miRNA–mRNA were involved in the network regulation map of muscle development in the three periods, which may indicate that they play an important role in the muscle growth and development of goats (Table 4).

In order to further understand the functional significance of lncRNAs in the ceRNA network, GO functional enrichment analysis of targeted mRNAs was performed in ceRNA (Figure 6a). It was found that target genes were significantly enriched in the processes of actin filament binding, actin binding, actin cytoskeleton formation, and myosin complex formation. *ABLIM2*, *CTNNA3,* and *VCL* are involved in actin filament and actin binding processes, *MYH13*, *TNNT1*, LOC102181426, LOC102181869, and *TNNI1* are involved in myosin complex and actin cytoskeleton processes. In addition, KEGG enrichment results showed that the same target mRNA was mainly enriched in the FoxO signaling pathway, TNF signaling pathway, MAPK signaling pathway, and GnRH signaling pathway (Figure 6b–d).

### 3.8. Verification of Differentially Expressed lncRNAs

To verify the accuracy of the RNA-seq results, we randomly selected four differentially expressed lncRNAs and detected their expression patterns at three different developmental stages by RT-qPCR. The RT-qPCR results confirmed that these four differentially expressed lncRNAs were expressed at different developmental stages (Figure 7), and the expression patterns of the differentially expressed lncRNAs were consistent with the RNA-seq results, indicating that the RNA-seq data were authentic and reliable.

## 4. Discussion

As the largest organ in the body, skeletal muscle accounts for about 40–50% of the body weight of mammals [31]. The growth and development of skeletal muscle is a prerequisite for the normal functioning of basics such as movement, respiration, and metabolism. Muscle fiber is the basic unit of muscle, and the characteristics and types of muscle fiber can be used as the index of meat quality evaluation. Muscle shear force is affected by muscle fiber diameter and cross-sectional area, and it is generally believed that the smaller the muscle fiber diameter and cross-sectional area, the smaller the muscle shear force, and the more tender the meat [32]. Meat color determines consumers’ desire to buy, pH value helps to maintain the flavor and taste of meat, the difference in meat color is mainly derived from the content of myoglobin, and age affects the concentration of muscle myoglobin, which increases with age [33]. According to the metabolic characteristics of muscles, muscle fibers can be divided into oxidized and glycolytic types. Muscles with oxidized fibers contain more myoglobin than those with glycolytic fibers, which is also one of the reasons for the different meat color [34,35]. The extent to which muscle pH decreases after death depends on the content of muscle glycogen. When animals are slaughtered, the glycogen in muscle will be converted into lactic acid, resulting in a decrease in pH value [36]. Based on the contraction characteristics of the muscle, muscle fibers can be divided into fast muscle and slow muscle. Fast muscle fibers generally have a lower pH than slow muscle due to their higher glycogen content [37]. In this study, by analyzing the muscle fiber characteristics, we found that the muscle fiber diameter and area of 2-month-old goats were significantly smaller than those of 6-month-old and 10-month-old goats, which was consistent with the result that the shear force of 2-month-old goats was significantly smaller than those of 6-month-old and 10-month-old goats. The lightness of 10-month-old goats was the smallest of the three ages of goats, which may be related to the fact that the muscles of older animals have lower reflectivity [38]. The redness of 6-month-old goats was the largest among the three ages of goats, but the 10-month-old goats contained more myoglobin, which may be affected by the muscle pH value, which greatly impacts the stability of myoglobin reoxidation. The conformation of myoglobin changes at a low pH value meant the muscle was more easily oxidized and the meat color changed [39]. There was no obvious trend in the influence of age on yellowness, and some studies have proven that there is a greater relationship with the content of intramuscular fat [40]. The pH_24h_ of 10-month-old goats was the lowest, which may be related to the higher level of glycogen in pre-slaughter muscle. In addition, the rate and degree of decline in pH were significantly correlated with the water-holding capacity of the muscle, with rapid decreases in pH reducing the net charge of proteins, which in turn reduces the water-holding capacity of the muscle [41]. In these results, there was no significant difference in the drip loss of goat muscle between the three ages. Studies have shown that a final pH value lower (pH < 5.4) than the normal value in meat arises when muscle protein has the same amount of positive and negative charge, and the muscle has the lowest water-holding capacity at this time [42]. The pH values of the three ages in this study were different but still within the normal pH range.

In this study, we identified a total of 3073 lncRNAs in the *M. Longissimus dorsi* of goats at three different growth periods, including 2676 known lncRNAs and 397 predicted novel lncRNAs. Among the screened lncRNAs, lincRNA was the most common, which was consistent with the results of Zhan [43], Gu [44], and Yang [45]. The identified lincRNAs had the characteristics of a shorter length, fewer exons, and shorter ORF length than the encoding transcript, which is consistent with the results of previous studies [46,47]. In addition, in order to verify the accuracy of the sequencing results, we applied strict sequencing data filtering and found that the RT-qPCR results were consistent with the sequencing results, proving that the sequencing results were true and credible.

Skeletal muscle growth and development form a complex process involving a large number of genes and regulatory factors, with coordinated interactions among various components. In order to understand the dynamics of skeletal muscle lncRNAs during goat growth and development, a total of 232 differentially expressed lncRNAs were identified, by whole-transcriptome sequencing, which may play important roles in muscle growth and development. Previous studies have reported the regulatory effect of lncRNA in skeletal muscle. For example, CTTN-IT1 regulates the expression of *YAP1* by competitively binding to miR-29a, thus promoting the proliferation and differentiation of skeletal muscle satellite cells [48]. lncIRS1 can act as a molecular sponge of the miR-15 family to regulate the expression of *IRS1*, promote muscle generation in skeletal muscle, and control muscle atrophy [49]. In the results of this paper, CREB5-IT1 and CREB5-IT3 were found to be differentially expressed in different goat longissimus dorsi tissues, which may further affect the expression of *CREB5*, leading to the occurrence of muscle weakness [50]. Therefore, the lncRNAs identified in this study may serve as important regulators of skeletal muscle development in goats.

Unlike proteins, the functions of lncRNAs cannot be predicted by the sequence or structure at present. Understanding the localization and local interactions of lncRNAs is the key to predicting their function [51]. Therefore, we predicted the function of lncRNAs by their positional relationship and expression correlation with protein-coding genes. Cis-regulatory refers to lncRNAs acting on nearby target genes [52]. When searching for potential protein-coding genes within 100 kb upstream and downstream of different lncRNAs, we found that some genes, such as *MKL2* [53], *RUFY1* [54], and *CREB5* [55], have been confirmed to play an important role in muscle development. This implies that lncRNAs can indirectly regulate muscle growth and development by regulating nearby protein-coding genes. In addition, lncRNAs can also function in a trans-regulatory manner, which mainly refers to the way in which lncRNAs affect the expression of target genes by interacting with trans-acting factors [56]. In our study, the trans-action of lncRNA was explored with the absolute value of the correlation coefficient of gene expression greater than 0.95 as the screening condition, and 1378 lncRNAs transcripts associated with protein-coding genes were detected. The corresponding coding genes of these lncRNA transcripts included *MYH14*, *MYH15*, *MYOG*, and *MYOC*. He [57] studied the lncRNA-specific expression profile of longissimus dorsi development in the Tianzhu white yak and found that *MYH14* was also involved in the process of muscle development. Zhang [58] found that *MYH15* was involved as a regulatory factor in the growth of three chicken mammary muscles. In summary, lncRNA can participate in the regulation of muscle growth and development through cis or trans mechanisms of action.

Previous studies have shown that the expression of target genes can be indirectly regulated through the molecular sponging action of lncRNA. Therefore, in this paper, the role of lncRNA in muscle growth and development was analyzed by constructing the lncRNA–miRNA–mRNA networks. We predict that in these networks, CREB5-IT3, 108634775-AS1, and 108635404-AS2 can act as molecular sponges for chi-miR-127-5p, and that CREB5-IT3 and XR_001297299.2 act as molecular sponges for chi-miR-758 to regulate muscle growth and development by modulating the target genes of these miRNAs. It has been demonstrated that the expression of miR-127-5p is associated with the translational efficiency of β-F1-ATPase, which further affects muscle protein metabolism and ultimately contributes to the development of obesity [59]. In yak rump muscle fat deposition, miR-127-5p is also one of the core components of the miRNA–mRNA network [60]. Furthermore, the expression of chi-miR-758 inhibits *PPP2R2C* and promotes the differentiation of bovine skeletal muscle myoblasts [61]. Among the ceRNA networks studied in this paper, chi-miR-485-5p [62], miR-15b-5p [63], and miR-497-5p [64] have been reported in other studies to be involved in regulating the proliferation and differentiation of myoblasts. We analyzed the role of target genes in the ceRNA network by GO functional enrichment and KEGG pathway enrichment, and the results were significantly enriched in terms such as actin binding and the actin backbone, demonstrating that some lncRNAs are involved in the growth and development of goat skeletal muscle at different developmental periods. The KEGG pathway is significantly enriched in signaling pathways such as MAPK, FoxO, and GnRH, which have been proven to be key pathways regulating muscle growth and development [65,66,67].

## 5. Conclusions

In conclusion, we first described the differences in the muscle fiber and meat quality of YDW goats at different ages and determined that muscle fiber characteristics influence meat quality. Second, we outlined the lncRNA expression profile during the development of the longissimus dorsi muscle in YDW goats at 2, 6, and 10 months of age. It was found that the target genes of lncRNAs were significantly enriched in AMPK, FoxO, GnRH, and other pathways closely related to muscle growth and development. Additionally, in the three ceRNA networks, CREB5-IT3, XR_001297299.2, 108634775-AS1, and 108635404-AS2 were identified as being involved in the growth and development of goat muscle during the three periods, potentially providing important directions for future research. In summary, this study has identified several candidate lncRNAs with significant regulatory effects on skeletal muscle development in goats, offering valuable references for future research on muscle development in goats.

## Figures and Tables

**Figure 1 animals-14-03125-f001:**
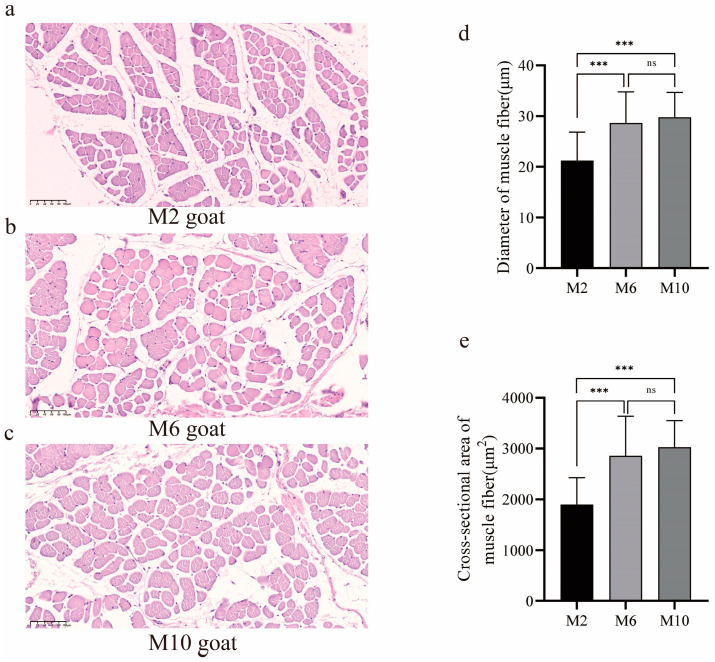
The difference in muscle fibers of *m. longissimus dorsi* of YDW goats at different ages. (**a**) Muscle fiber sections of goats aged 2 months; (**b**) muscle fiber sections of goats aged 6 months; (**c**) muscle fiber sections of goats aged 10 months; (**d**) comparison of muscle fiber diameter in goats of different months of age; (**e**) comparison of cross-sectional area of muscle fiber in goats of different months of age. ns, no difference, *** *p* < 0.01.

**Figure 2 animals-14-03125-f002:**
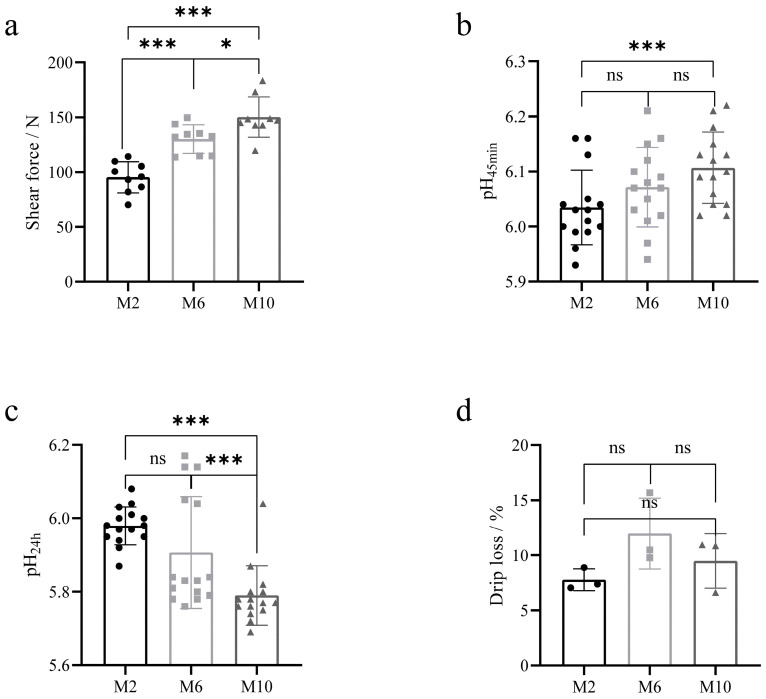
Comparison of meat quality levels of goats at different months. (**a**) Comparative analysis of muscle shear force; (**b**) comparative analysis of muscle pH_45min_; (**c**) comparative analysis of muscle pH_24h_; (**d**) comparative analysis of muscle drip loss. * *p* < 0.05; *** *p* < 0.01. The circle represents the data of each index of the goat at 2 months of age, the square represents the data of each index of the goat at 6 months of age, and the triangle represents the data of each index of the goat at 10 months of age.

**Figure 3 animals-14-03125-f003:**
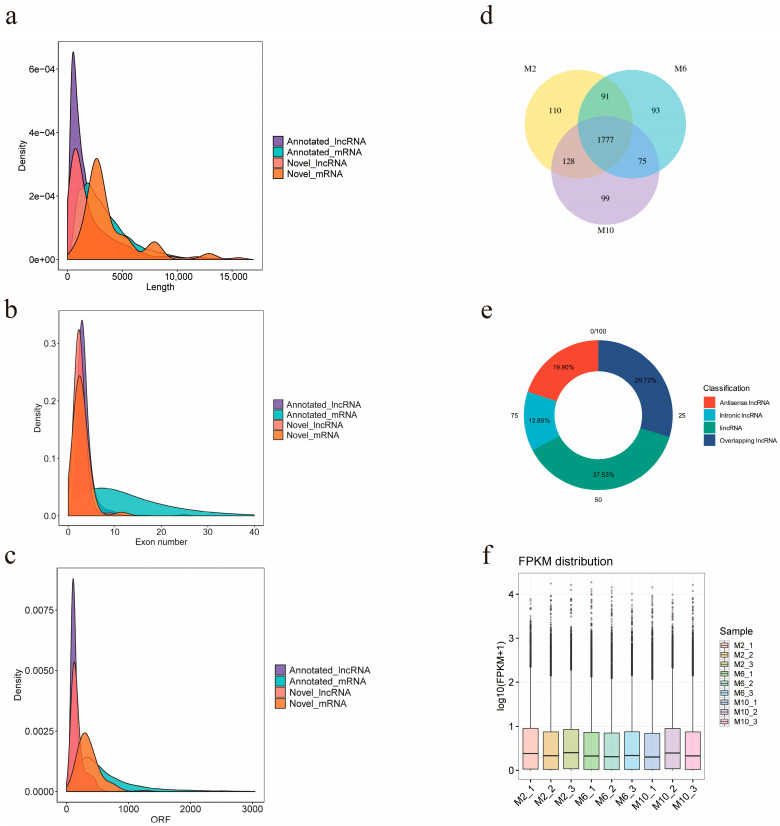
Prediction and classification of lncRNA of YDW goats. (**a**) Length comparative density distribution map of lncRNA and mRNA; (**b**) exon number density distribution map of lncRNA and mRNA; (**c**) ORF length density distribution of lncRNA and mRNA; (**d**) Venn diagram of lncRNA expression in muscles of goats of different months of age; (**e**) lncRNA type distribution map; (**f**) expression level distribution map of each sample.

**Figure 4 animals-14-03125-f004:**
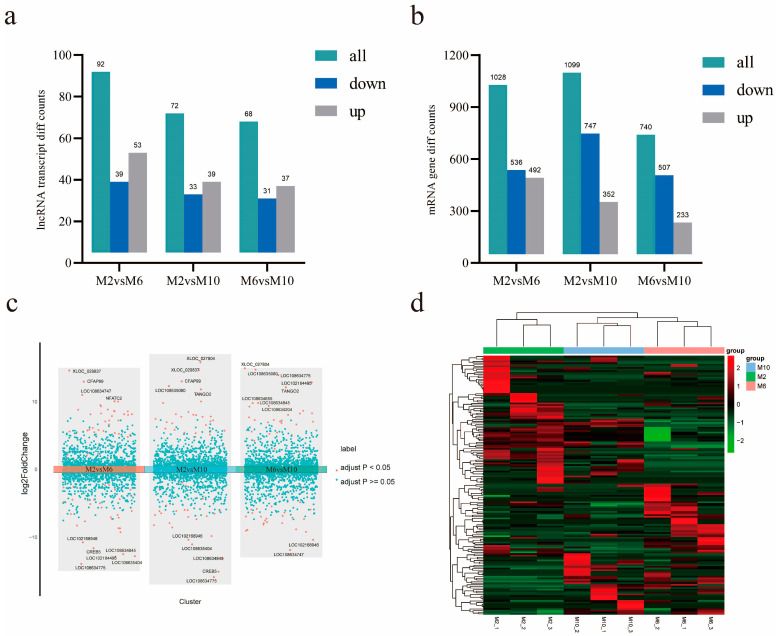
Analysis of mRNA and lncRNA differential expression. (**a**) LncRNA differential expression in goats of different months; (**b**) mRNA differential expression in goats of different months; (**c**) comparison of different volcanic maps of goats in different groups; (**d**) expression heat maps of different lncRNAs in samples of different months of age; the line is lncRNA, column is tissue.

**Figure 5 animals-14-03125-f005:**
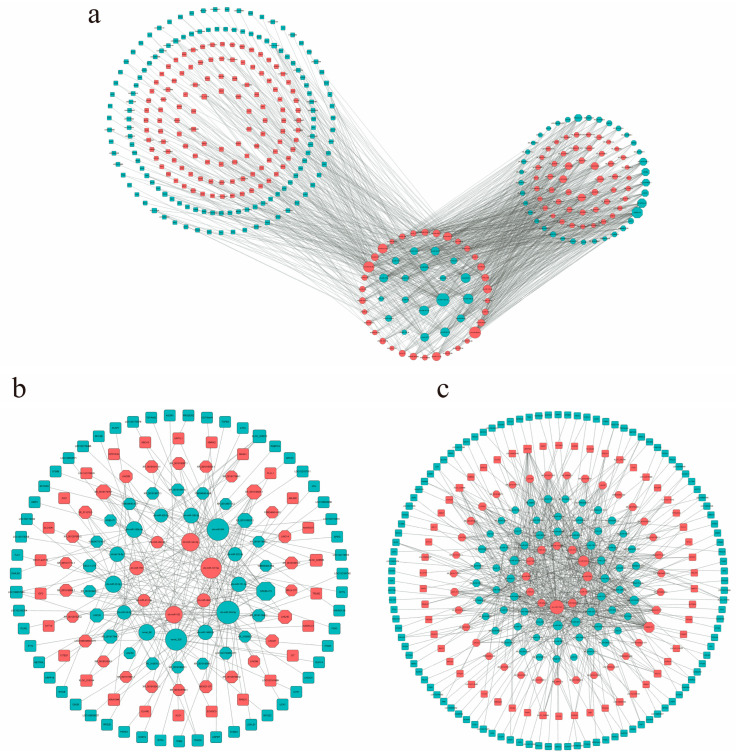
LncRNA-related ceRNA networks. (**a**) M2vsM6ceRNA network; (**b**) M2vsM10ceRNA network; (**c**) M6vsM10ceRNA network. Different colors represent the up–down relationship: red means up, blue means down. The size of a node represents the degree of this node. The greater the degree, the more edges are connected to this node, and the more important the node is in the network.

**Figure 6 animals-14-03125-f006:**
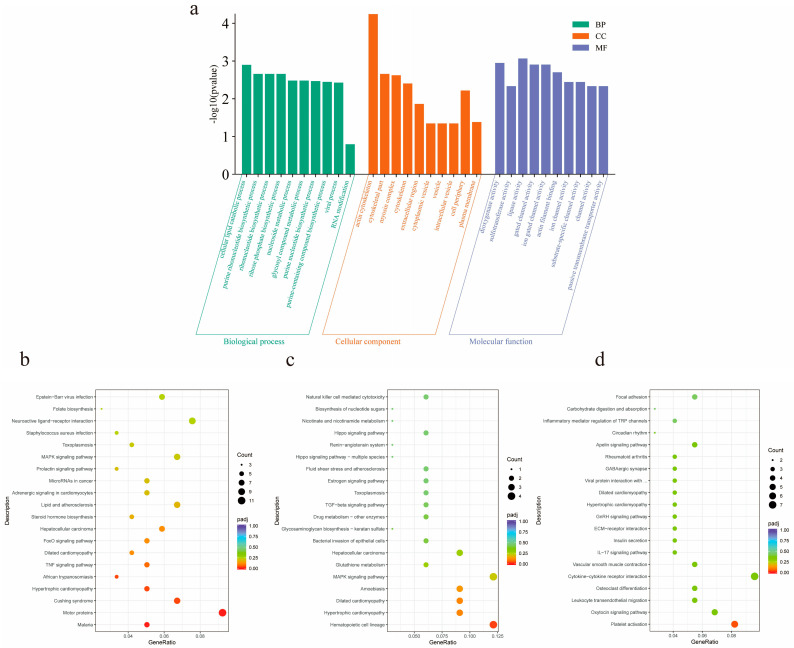
Functional analysis of differential lncRNA target genes. (**a**) The first 30 GO terms enriched for differential lncRNA target genes in three groups of pairwise comparisons of goats of different months of age. Green represents biological processes, orange represents cellular components, and purple represents molecular functions; (**b**) the top 20 KEGG pathways of M2vsM6 differential lncRNA target genes; (**c**) the top 20 KEGG pathways of M2vsM10 differential lncRNA target genes; (**d**) the top 20 KEGG pathways of M6vsM10 differential lncRNA target genes.

**Figure 7 animals-14-03125-f007:**
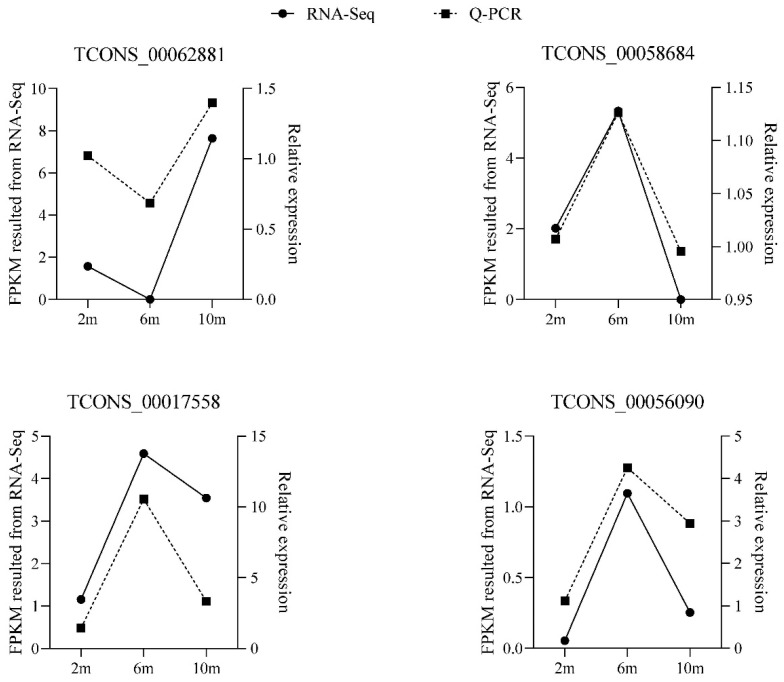
RT-qPCR validation of differentially expressed lncRNAs.

**Table 1 animals-14-03125-t001:** Sequence of qPCR primers used for validation.

Gene Name	Primer Type	Sequence of Primers
TCONS_00062881	Forward	ATCCGCAGCAGGTCTCCAAG
	Reverse	AGCCCTTAGAGCCAATCCTTATCC
TCONS_00058684	Forward	CCTGCGTCCTCTCCTCTTGG
	Reverse	CGGATGAAGCGGAACGGATATG
TCONS_00017558	Forward	TTCCTCAGCCCACTCCAAACAAAC
	Reverse	GCGAGCACAGAGCCCAAACC
TCONS_00056090	Forward	TAGAGGAAGGAAGCGGGAAGGAAG
	Reverse	TCGGTTTAGCCAACAGGGTCTTTG
ACTB	Forward	AGCCTTCCTTCCTGGGCATGGA
	Reverse	GGACAGCACCGTGTTGGCGTAA

**Table 2 animals-14-03125-t002:** Quality of YDW goat meat at different ages.

Item	Group			
	2 m	6 m	10 m	*p*-Value
Lightness (L*)	43.66 ± 4.44 ^a^	44.61 ± 4.52 ^a^	35.08 ± 1.15 ^b^	<0.01
Redness (a*)	13.12 ± 3.48 ^a^	19.11 ± 6.25 ^b^	4.93 ± 1.01 ^c^	<0.01
Yellowness (b*)	8.69 ± 2.94 ^b^	16.10 ± 3.58 ^a^	10.23 ± 0.73 ^b^	<0.01
pH_45min_	6.03 ± 0.068 ^b^	6.07 ± 0.072 ^ab^	6.11 ± 0.065 ^a^	0.006
pH_24h_	5.98 ± 0.051 ^a^	5.91 ± 0.152 ^a^	5.79 ± 0.081 ^b^	<0.01
Shear force (N)	98.68 ± 10.70 ^a^	130.06 ± 13.08 ^b^	150.10 ± 18.44 ^c^	<0.05
Drip loss%	7.8 ± 1.0	12.0 ± 3.2	9.5 ± 2.5	>0.05

Note: 2 m, 6 m, and 10 m in the table represent 2-month-old, 6-month-old, and 10-month-old YDW goats, respectively. The same row of data in the same index with the same superscript letters (a, b, c) indicates non-significant differences, while the data with different letters indicate significant differences. *p*-values indicate the muscle quality levels of goats at different months of age.

**Table 3 animals-14-03125-t003:** Overview of RNA-seq results for each sample.

Sample Name	Raw Reads	Clean Reads	Q20 (%)	Q30 (%)	Total Mapping (%)
M2-1	91,139,898	89,638,430	98.46	95.95	86,129,394 (96.09%)
M2-2	87,557,804	86,071,772	98.11	95.22	82,055,643 (95.33%)
M2-3	83,098,176	81,743,532	98.03	95.12	77,764,683 (95.13%)
M6-1	86,928,100	85,492,412	98.03	95.13	81,247,096 (95.03%)
M6-2	89,144,180	87,765,748	97.99	95.01	83,546,691 (95.19%)
M6-3	87,391,674	86,154,582	98.07	95.17	81,960,778 (95.13%)
M10-1	82,018,490	80,757,046	98.21	95.45	77,024,837 (95.38%)
M10-2	89,452,288	88,011,694	98.03	95.06	83,682,629 (95.08%)
M10-3	89,534,296	88,120,994	97.94	94.94	83,555,700 (94.82%)

**Table 4 animals-14-03125-t004:** Prediction of important ceRNA network regulation maps.

lncRNA	miRNA	mRNA
CREB5-IT3	chi-miR-758	WNT2
XR_001297299.2	chi-miR-758	ADCY2
CREB5-IT3	chi-miR-758	LRRTM3
CREB5-IT3	chi-miR-127-5p	OPTN
108635404-AS2	chi-miR-127-5p	CITED1
108634775-AS1	chi-miR-127-5p	S100A1

## Data Availability

The raw sequence data reported in this paper have been deposited in the Genome Sequence Archive (Genomics, Proteomics & Bioinformatics 2024) in the National Genomics Data Center (Nucleic Acids Res 2024), China National Center for Bioinformation/Beijing Institute of Genomics, Chinese Academy of Sciences (CRA018697), and they are publicly accessible at https://ngdc.cncb.ac.cn/gsa (accessed on 10 April 2024).

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
