# Peer review of "Analysis of lncRNAs and Their Regulatory Network in Skeletal Muscle Development of the Yangtze River Delta White Goat"

_animals, 2024, doi:10.3390/ani14213125_

Round 1
Reviewer 1 Report
Comments and Suggestions for Authors
Submitted paper, entitled "Analysis of lncRNAs and Their Regulatory Network in Skeletal Muscle Development of Yangze River Delta White Goat" in my opinion is very interesting and introduces new and valuable data into state of current knowledge. The paper is good prepared, applied analytical methods and bioinformatics tools are well chosen and enough described. I think that results are clear presented and good supported by numerous figures and tables; they are good and comprehensive discussed too.
However, there are some minor points that needs correction before publishing in Animals. See comments below:
line 16 - delete " identified lncRNAs specifically expressed at each growth stage"
line 24 - explain ceRNA shortcut
lines 27, 46, 87 and rest - italicize latin name of muscle and add m.- m. longissimus dorsi
line 53 - explain circRNAs and miRNAs shorcuts (lncRNAs is explained in line 22)
line 54 - "might play an important role in coding gene expression and regulation of coding genes" → I propose "might play an important role in protein-coding gene expression and its regulation"
line 84 - "small stink" better "low odour"
line 90 - delete sentence "In this study, we compared the lncRNA transcripts of longissimus dorsi of YDW goats at three stages, and then constructed and analyzed lncRNA associated ceRNA networks." - repeated data
line 102 - "The longissimus dorsi..." add "After slaughter, the m. longissimus dorsi..."
line 103 - delete " the samples"
line 239 - explain HE shortcut
line 249 - delete " *P < 0.05" (not visible in pictures above)
line 254 - add "...at 2 and 10 months of age..." and delete " but there were no significant differences with those of goats at 2 months of age (P > 0.05)"
Table 2 - delete "a" letter from drip loss values (no need to indicate when values not differ)
line 293 - add "respectively" at the end
line 349 and Table 4 - "lnRNA" → "lncRNA"
Lines 372-374, 452, 455, 458, 467 - italicize gene symbols
Line 414 - "In this study, we found by analyzing the characteristics of muscle fiber diameter and muscle fiber area: The muscle fiber diameter and muscle fiber area of 2-month-old goats were significantly smaller..." I propose "In this study, by analyzing the muscle fiber characteristics we found that the muscle fiber diameter and area of 2-month-old goats were significantly smaller ..."
If it is possible give pictures 4C; 5 A, B, C; 6 B, C, D with higher resolution!!!
Author Response
Thank you for spending your value time on our manuscript “Analysis of lncRNAs and Their Regulatory Network in Skeletal Muscle Development of Yangze River Delta White Goat” (ID: animals-3258947). Thank you for your comments. We have gone over all of comments, and the revised text is shown in red in the revised manuscript. I hope our revision has addressed all the comments from you. The specific changes in the revised paper are listed below.
Comment 1: delete " identified lncRNAs specifically expressed at each growth stage"
Response 1: Thank you for your valuable suggestion. We agree with this comment. We have deleted "identified lncRNAs specifically expressed at each growth stage". More details are shown below:
Line 15-21: In this study, we identified potential new lncRNAs, described lncRNAs expression profiles at different developmental stages of Yangze River Delta White (YDW) goat, and constructed corresponding ceRNA regulatory networks. This result provides valuable genomic data for studying the growth and development process of animals, and may provide theoretical basis for studying the biological functions of lncRNAs in animal growth and development. We've already revised it in line 15-21 in the revised manuscript
Comment 2: explain ceRNA shortcut
Response 2: Thank you for your valuable suggestion. We agree with this comment. We explained ceRNA shortcut. Details as follows:
Line 23-26: This paper aims to explore the potential role and dynamic change process of lncRNAs and related ceRNA (competitive endogenous RNA) networks in skeletal muscle development of Yangze River Delta White (YDW) goat, and to analyze the differences in muscle fibre characteristics and meat quality of goats at different growth stages. We've already revised it in line 23-26 in the revised manuscript.
Comment 3: italicize latin name of muscle and add m.- m. longissimus dorsi
Response 3: Thank you for your valuable suggestion. We agree with this comment. We have italicized the Latin names of muscles and add m.- m. longissimus dorsi. Details as follows:
Line 27-28: we compared the expression profiles of lncRNAs in the m. longissimus dorsi of the YDW goats at different stages of growth and development by RNA sequencing. We've already revised it in line 27-28 in the revised manuscript.
Line 47-48: Yangze River Delta White (YDW) goat; lncRNAs; M. Longissimus dorsi; Meat quality; Muscle fiber characteristics; ceRNA network. We've already revised it in line 47-48 in the revised manuscript.
Line 89-90: we used high-throughput sequencing technologies to analyze the transcriptome of the m. longissimus dorsi of YDW goats from different periods. We've already revised it in line 89-90 in the revised manuscript.
Line 104-105: After slaughter, the m. longissimus dorsi samples were collected between the 12th and 13th thoracic vertebrae of each goat. We've already revised it in line 104-105 in the revised manuscript.
Line 112-113: The collected samples of the m. longissimus dorsi were im-mersed and fixed in 4% paraformaldehyde. We've already revised it in line 112-113 in the revised manuscript.
Line 148-149: Total RNA was extracted from m. longissimus dorsi of goats by Trizol (Tiangen, Beijing, China) method. We've already revised it in line 148-149 in the revised manuscript.
Line 228-229: Total RNA was extracted from m. longissimus dorsi of goats by Trizol (Tiangen, Beijing, China). We've already revised it in line 228-229 in the revised manuscript.
Line 247-248: HE (Hematoxylin-Eosin staining) sections of the m. longissimus dorsi revealed differences in the muscle fibers of YDW goats at different ages. We've already revised it in line 247-248 in the revised manuscript.
Line 254-255: The difference of muscle fibers of m. longissimus dorsi of YDW goats at different ages. We've already revised it in line 254-255 in the revised manuscript.
Line 448-449: we identified a total of 3073 lncRNAs in the m. longissimus dorsi of goats at three different growth periods. We've already revised it in line 448-449 in the revised manuscript.
Line 467-469: CREB5-IT1 and CREB5-IT3 were found to be differentially expressed in different goat m. longissimus dorsi tissues. We've already revised it in line 467-469 in the revised manuscript.
Line 487-489: studied that the lncRNAs specific expression profile of m. longissimus dorsi development in Tianzhu white yak and found that MYH14 was also in-volved in the process of muscle development. We've already revised it in line 487-489 in the revised manuscript.
Line 518-520: we outlined the lncRNA expression profile during the development of the m. longissimus dorsi muscle in YDW goats at 2, 6, and 10 months of age. We've already revised it in line 518-520 in the revised manuscript.
Comment 4: explain circRNAs and miRNAs shorcuts (lncRNAs is explained in line 22)
Response 4: Thank you for your valuable suggestion. We agree with this comment. We explain circRNAs and miRNAs shortcuts. Details as follows:
Line 52-56: However, with the development of transcriptome sequencing technology, more and more non-coding RNAs, including lncRNAs, circRNAs (circular RNA), miRNAs (microRNA) were unearthed, and it was found that these so-called non-coding RNAs might play an important role in protein-coding gene expression and its regulation. We've already revised it in line 52-56 in the revised manuscript.
Comment 5: "might play an important role in coding gene expression and regulation of coding genes" → I propose "might play an important role in protein-coding gene expression and its regulation"
Response 5: Thank you for your valuable suggestion. We agree with this comment. We have finished revising this sentence. Details as follows:
Line 52-57: However, with the development of transcriptome sequencing technology, more and more non-coding RNAs, including lncRNAs, circRNAs (circular RNA), miRNAs (microRNA) were unearthed, and it was found that these so-called non-coding RNAs might play an important role in protein-coding gene expression and its regulation We've already revised it in line 52-57 in the revised manuscript.
Comment 6: "small stink" better "low odour"
Response 6: Thank you for your valuable suggestion. We agree with this comment. We have revised the "small stink" to "low odour". Details as follows:
Line 85-88: The Yangtze River Delta White (YDW) goat is an excellent native goat breed in China, which has excellent characteristics such as low odour, tender meat and even fat distribution, and is also the only goat breed in China that can produce high-quality penned wool with both excellent skin, meat and wool. We've already revised it in line 85-88 in the revised manuscript.
Comment 7: delete sentence "In this study, we compared the lncRNA transcripts of longissimus dorsi of YDW goats at three stages, and then constructed and analyzed lncRNA associated ceRNA networks." - repeated data
Response 7: Thank you for your valuable suggestion. We agree with this comment. We have deleted "In this study, we compared the lncRNA transcripts of longissimus dorsi of YDW goats at three stages, and then constructed and analyzed lncRNA associated ceRNA networks." Details as follows:
Line 88-97: In this paper, we used high-throughput sequencing technologies to analyze the transcriptome of the m. longissimus dorsi of YDW goats from different periods, to understand the dynamic change process of lncRNAs in the skeletal muscles of goats, to predict the potential functions of lncRNAs, and to construct ceRNA networks related to the analyzed lncRNAs. These results can help in understanding the differences in muscle fiber development and muscle quality of YDW goats at different developmental stages, as well as the molecular mechanisms that regulate skeletal muscle development throughout the life cycle. We've already revised it in line 88-97 in the revised manuscript.
Comment 8: "The longissimus dorsi..." add "After slaughter, the m. longissimus dorsi..."
Response 8: Thank you for your valuable suggestion. We agree with this comment. We have added "After slaughter, the m. longissimus dorsi..." Details as follows:
Line 104-105: After slaughter, the m. longissimus dorsi samples were collected between the 12th and 13th thoracic vertebrae of each goat. We've already revised it in line 104-105 in the revised manuscript.
Comment 9: delete " the samples"
Response 9: Thank you for your valuable suggestion. We agree with this comment. We have deleted " the samples". Details as follows:
Line 104-107: After slaughter, the m. longissimus dorsi samples were collected between the 12th and 13th thoracic vertebrae of each goat, and were immediately placed in liquid nitrogen for rapid freezing and transferred to a -80°C refrigerator for storage. We've already revised it in line 104-107 in the revised manuscript.
Comment 10: explain HE shortcut
Response 10: Thank you for your valuable suggestion. We agree with this comment. We explain HE shortcut. Details as follows:
Line 247-248: HE (Hematoxylin-Eosin staining) sections of the m. longissimus dorsi revealed differences in the muscle fibers of YDW goats at different ages. We've already revised it in line 247-248 in the revised manuscript.
Comment 11: delete " *P < 0.05" (not visible in pictures above)
Response 11: Thank you for your valuable suggestion. We agree with this comment. We have deleted " *P < 0.05". Details as follows:
Line 258: ns means no difference, ***P < 0.01. We've already revised it in line 258 in the revised manuscript.
Comment 12: add "...at 2 and 10 months of age..." and delete " but there were no significant differences with those of goats at 2 months of age (P > 0.05)"
Response 12: Thank you for your valuable suggestion. We agree with this comment. We have added "...at 2 and 10 months of age..." and deleted " but there were no significant differences with those of goats at 2 months of age (P > 0.05)" Details as follows:
Line 262-265: and L* values of the muscles of goats at 2 and 6 months of age were significantly greater than those of goats at 10 months of age (P < 0.01). We've already revised it in line 262-265 in the revised manuscript.
Comment 13: delete "a" letter from drip loss values (no need to indicate when values not differ)
Response 13: Thank you for your valuable suggestion. We agree with this comment. We have deleted "a" letter from drip loss values. Details as follows:
Line 275: Table 2, drip loss values. We've already revised it in line 275 in the revised manuscript.
Comment 14: add "respectively" at the end
Response 14: Thank you for your valuable suggestion. We agree with this comment. We have added "respectively" at the end. Details as follows:
Line 302-304: The screened lncRNAs were respectively classified into four types, with the percentage of lincRNA (37.53%), antisense lncRNA (19.90%), overlapping lncRNA (29.72%), and intronic lncRNA (12.85%). We've already revised it in line 302-304 in the revised manuscript.
Comment 15: "lnRNA" → "lncRNA"
Response 15: Thank you for your valuable suggestion. We agree with this comment. We have modified "lnRNA" → "lncRNA". Details as follows:
Line 372: Table 4, lncRNA. We've already revised it in line 372 in the revised manuscript.
Comment 16: Lines 372-374, 452, 455, 458, 467 - italicize gene symbols
Response 16: Thank you for your valuable suggestion. We agree with this comment. We have italicized gene symbols of Lines 372-374, 452, 455, 458, 467. Details as follows:
Line 381-383, 463, 466, 469, 478. We've already revised it in line 381-383, 463, 466, 469, 478 in the revised manuscript.
Comment 17: "In this study, we found by analyzing the characteristics of muscle fiber diameter and muscle fiber area: The muscle fiber diameter and muscle fiber area of 2-month-old goats were significantly smaller..." I propose "In this study, by analyzing the muscle fiber characteristics we found that the muscle fiber diameter and area of 2-month-old goats were significantly smaller ..."
Response 17: Thank you for your valuable suggestion. We agree with this comment. We have revised the sentence. Details as follows:
Line 423-425: In this study, by analyzing the muscle fiber characteristics we found that the muscle fiber diameter and area of 2-month-old goats were significantly smaller than those of 6-month-old and 10-month-old goats, which was consistent with the result that the shear force of 2-month-old goats was significantly smaller than that of 6-month-old and 10-month-old goats. We've already revised it in line 423-425 in the revised manuscript.
Comment 18: If it is possible give pictures 4C; 5 A, B, C; 6 B, C, D with higher resolution
Response 18: Thank you for your valuable suggestion. We agree with this comment.
We have provided pictures with higher resolution. Details please see the attachment.

Reviewer 2 Report
Comments and Suggestions for Authors
Dear authors, in section 2.1, you claim that there is no difference in the weight and height of the goats in the three groups! This should be supported by data from weight and height measurements that you are supposed to have! Please insert these data in table! The data in this supplemental table would better orient readers to the quantitative and qualitative changes in muscle development in growing goats. Please it would be better to note the sex of the goats, regardless of the fact that from the text to a certain extent it is understood that it is a question of female animals.
Author Response
Thank you for spending your value time on our manuscript “Analysis of lncRNAs and Their Regulatory Network in Skeletal Muscle Development of Yangze River Delta White Goat” (ID: animals-3258947). Thank you for your comments. We have gone over all of comments, and the revised text is shown in red in the revised manuscript. I hope our revision has addressed all the comments from you. The specific changes in the revised paper are listed below.
Comment 1: Dear authors, in section 2.1, you claim that there is no difference in the weight and height of the goats in the three groups! This should be supported by data from weight and height measurements that you are supposed to have! Please insert these data in table! The data in this supplemental table would better orient readers to the quantitative and qualitative changes in muscle development in growing goats. Please it would be better to note the sex of the goats, regardless of the fact that from the text to a certain extent it is understood that it is a question of female animals.
Response 1: Thank you for your valuable suggestion, this is our misrepresentation, we are very sorry for this error. What we mean is that goats of the same age are similar in weight and body condition, not that there is no difference in weight and height among the three groups. Moreover, we are sorry for our carelessness, all the goats in this paper are male. Details as follows:
Line 100-101: The 9 healthy YDW male goats involved in this experiment were all from the goat breeding farm in Lishui City, Zhejiang Province, China. We've already revised it in line 100-101 in the revised manuscript.
Line 101-103: Three goats at 2, 6, and 10 months of age were selected, and the goats of the same age are similar in weight and body condition. We've already revised it in line 101-103 in the revised manuscript.
Reviewer 3 Report
Comments and Suggestions for Authors
In this manuscript, authors compared the lncRNA transcripts of longissimus dorsi of YDW goats at three stages, and then constructed and analyzed lncRNA associated ceRNA networks. These results can help in understanding the differences in muscle fiber development. Before the manuscript can be accepted, many doubts need to be clarified. More details see below:
L15-18. ‘In this study, we identified potential new lncRNAs, identified lncRNAs specifically expressed at each growth stage, described lncRNAs expression profiles at different developmental stages of Yangze River Delta White (YDW) goat, and constructed corresponding ceRNA regulatory networks.’
First, for those new lncRNA, please provide the info for these new lncRNAs. Meanwhile, the authors identified the goat lncRNAs of three growth stages, which could not represent "at each growth stage".
L86-95. Line 86-89 and Line 90-92 are duplicated.
L100-101. ‘There were no significant differences in the height and weight of the goats among all groups’.
No difference here means no difference within the groups, or no difference between the groups. IF there was no difference in height and weight between different groups, according to common sense, this seems unlikely, and there is no need for the authors to deliberately keep the sample consistent in height and weight, because the comparison is between different age groups, and it is normal for them to have differences in height and weight.
L182. gffcompare software, software version is required.
L183-184. ‘Finally, transcripts with no coding potential from the three databases results of CPC2 [25], PFAM [26] and CNCI [27] were selected as the candidate novel lncRNAs data set we predicted.’
Here the author does not explain how the results of the three databases are handled, and whether they are union or intersect;
L190. P<0.05. According to Fig4c, it was adjust P < 0.05, not p <0.05, that was different. Therefore, it should be adjust P in line 190.
L191. |log2FoldChange| ≥ 0. If it was true, here ‘fold change’ would be 1, that means case/control = 1, there would be no changes between two groups, and the following results would be meaningless. Please check or give a full explanation.
L220-230. The abbreviations of quantitative real-time PCR in the whole text are inconsistent, such as qRT-PCR, RT-PCR, qPCR.
L256. ’pH45min’ , ‘pH24h’, or ‘pH10h’, please keep consistent through the manuscript.
L326-344. ‘3.6. Prediction and enrichment analysis of lncRNA target genes’. This result description is not supported by any charts, tables or supplementary files, how can you ensure their reliability?
Others :
The English grammar of the whole text needs to be polished.
Comments on the Quality of English Language
The English grammar of the whole text needs to be polished.
Author Response
Thank you for spending your value time on our manuscript “Analysis of lncRNAs and Their Regulatory Network in Skeletal Muscle Development of Yangze River Delta White Goat” (ID: animals-3258947). Thank you for your comments. We have gone over all of comments, and the revised text is shown in red in the revised manuscript. I hope our revision has addressed all the comments from you. The specific changes in the revised paper are listed below.
Comment 1: line 15-18. ‘In this study, we identified potential new lncRNAs, identified lncRNAs specifically expressed at each growth stage, described lncRNAs expression profiles at different developmental stages of Yangze River Delta White (YDW) goat, and constructed corresponding ceRNA regulatory networks.’ First, for those new lncRNA, please provide the info for these new lncRNAs. Meanwhile, the authors identified the goat lncRNAs of three growth stages, which could not represent "at each growth stage".
Response 1: Thank you for the reminder. We agree with this comment. We are sorry for our carelessness. We will provide information on these new lncRNAs, moreover, Ambiguity due to description, We have deleted "identified lncRNAs specifically expressed at each growth stage". Details as follows:
Line 15-21: In this study, we identified potential new lncRNAs, described lncRNAs expression profiles at different developmental stages of Yangze River Delta White (YDW) goat, and constructed corresponding ceRNA regulatory networks. This result provides valuable genomic data for studying the growth and development process of animals, and may provide theoretical basis for studying the biological functions of lncRNAs in animal growth and development. We've already revised it in line 15-21 in the revised manuscript.
Comment 2: Line 86-89 and Line 90-92 are duplicated.
Response 2: Thank you for your valuable suggestion. We agree with this comment. We have deleted "In this study, we compared the lncRNA transcripts of longissimus dorsi of YDW goats at three stages, and then constructed and analyzed lncRNA associated ceRNA networks." Details as follows:
Line 88-97: In this paper, we used high-throughput sequencing technologies to analyze the transcriptome of the m. longissimus dorsi of YDW goats from different periods, to understand the dynamic change process of lncRNAs in the skeletal muscles of goats, to predict the potential functions of lncRNAs, and to construct ceRNA networks related to the analyzed lncRNAs. These results can help in understanding the differences in muscle fiber development and muscle quality of YDW goats at different developmental stages, as well as the molecular mechanisms that regulate skeletal muscle development throughout the life cycle. We've already revised it in line 88-97 in the revised manuscript.
Comment 3: Line 100-101. ‘There were no significant differences in the height and weight of the goats among all groups.’ No difference here means no difference within the groups, or no difference between the groups. IF there was no difference in height and weight between different groups, according to common sense, this seems unlikely, and there is no need for the authors to deliberately keep the sample consistent in height and weight, because the comparison is between different age groups, and it is normal for them to have differences in height and weight.
Response 3: Thank you for your valuable suggestion, this is our misrepresentation, we are very sorry for this error. What we mean is that goats of the same age are similar in weight and body condition, not that there is no difference in weight and height among the three groups. Details as follows:
Line 101-103: Three goats at 2, 6, and 10 months of age were selected, and the goats of the same age are similar in weight and body condition. We've already revised it in line 101-103 in the revised manuscript.
Comment 4: Line 182. gffcompare software, software version is required
Response 4: Thank you for your valuable suggestion. We agree with this comment. We have added gffcompare software version (version 0.10.6). Details as follows:
Line 186-187: The transcripts are compared with known databases by gffcompare (version 0.10.6) software. We've already revised it in line 186-187 in the revised manuscript.
Comment 5: Line 183-184. ‘Finally, transcripts with no coding potential from the three databases results of CPC2 [25], PFAM [26] and CNCI [27] were selected as the candidate novel lncRNAs data set we predicted.’ Here the author does not explain how the results of the three databases are handled, and whether they are union or intersect;
Response 5: Thank you very much for your valuable suggestion. Whether a transcript has coding potential is a key condition for judging whether it is a lncRNA. For the transcripts obtained above, Combining the current mainstream coding potential analysis methods CPC2, PFAM, CNCI to predict coding potential, The intersection of transcripts without coding potential in these software results was selected as the candidate Novel_lncRNA data set predicted by this analysis. Details as follows:
Line 188-190: Finally, the intersection of transcripts without coding potential was selected from the three databases results of CPC2 [25], PFAM [26] and CNCI [27] as the candidate novel lncRNAs dataset for our prediction. We've already revised it in line 188-190 in the revised manuscript.
Comment 6: Line 190. P<0.05. According to Fig4c, it was adjust P < 0.05, not p <0.05, that was different. Therefore, it should be adjust P in line 190.
Response 6: Thank you for your valuable suggestion. We agree with this comment. We have modified "P value < 0.05" → "adjust P value < 0.05". Details as follows:
Line 197-199: The significantly differentially expressed genes or lncRNAs were screened with adjust P value < 0.05 and |log2FoldChange| ≥ 1 as the conditions. We've already revised it in line 197-199 in the revised manuscript.
Comment 7: Line 191. |log2FoldChange| ≥ 0. If it was true, here ‘fold change’ would be 1, that means case/control = 1, there would be no changes between two groups, and the following results would be meaningless. Please check or give a full explanation.
Response 7: Thank you for the reminder. We agree with this comment. We are sorry for our carelessness. We have revised ‘|log2FoldChange| ≥ 0’ → ‘|log2FoldChange| ≥ 1’. Details as follows:
Line 197-199: The significantly differentially expressed genes or lncRNAs were screened with adjust P value < 0.05 and |log2FoldChange| ≥ 1 as the conditions. We've already revised it in line 197-199 in the revised manuscript.
Comment 8: Line 220-230. The abbreviations of quantitative real-time PCR in the whole text are inconsistent, such as qRT-PCR, RT-PCR, qPCR.
Response 8: Thank you for your valuable suggestion. We agree with this comment.
We have already revised the abbreviations of quantitative real-time PCR to qPCR in line 228-237 in the revised manuscript
Comment 9: Line 256. ’pH45min’, ‘pH24h’, or ‘pH10h’, please keep consistent through the manuscript.
Response 9: Thank you for your valuable suggestion. We agree with this comment.
We have already revised the ‘pH45min’ ‘pH24h’ and keep consistent through the revised manuscript.
Comment 10: Line 326-344. ‘3.6. Prediction and enrichment analysis of lncRNA target genes.’ This result description is not supported by any charts, tables or supplementary files, how can you ensure their reliability?
Response 10: Thank you for your valuable suggestion. We agree with this comment. See the attachment for details supporting this result description
